# Ideal acoustic quantum spin Hall phase in a multi-topology platform

Xiao-Chen Sun[1,2,4], Hao Chen[1,4], Hua-Shan Lai[1,4], Chu-Hao Xia[1], Cheng He [1,2,3] ✉ & Yan-Feng Chen [1,2,3] ✉

Fermionic time-reversal symmetry ($T_f$)-protected quantum spin Hall (QSH) materials feature gapless helical edge states when adjacent to arbitrary trivial cladding materials. However, due to symmetry reduction at the boundary, bosonic counterparts usually exhibit gaps and thus require additional cladding crystals to maintain robustness, limiting their applications. In this study, we demonstrate an ideal acoustic QSH with gapless behaviour by constructing a global $T_f$ on both the bulk and the boundary based on bilayer structures. Consequently, a pair of helical edge states robustly winds several times in the first Brillouin zone when coupled to resonators, promising broadband topological slow waves. We further reveal that this ideal QSH phase behaves as a topological phase transition plane that bridges trivial and higher-order phases. Our versatile multi-topology platform sheds light on compact topological slow-wave and lasing devices.

Topological insulators[1-4] (TIs) originating from solid-state materials manifest insulating interiors but conducting surfaces when adjacent to topologically distinct materials, e.g., trivial free space. This concept has spread into classical wave systems[5-17] in the past decade. The gapless edge states of TIs supporting robust propagation against defects leads to many unique phenomena and fantastic applications, such as robust slow waves[18-20] and stable lasers[21,22]. A typical case is the family of quantum Hall effects. For example, photonic[5,6] and phononic[16,17] quantum anomalous Hall effects with broken time-reversal symmetry provide support for gapless chiral edge states at boundaries when adjacent to arbitrary trivial cladding layers, such as crystal structures[21], perfect electric conductors[6], complete reflection boundaries[23], and even radiation boundaries[24]. Nevertheless, these models necessitate external or equivalent magnetic fields, thus suffering from narrow bandwidths, low operating frequency windows, and magnetic application scenarios.

Magnetic-free quantum spin Hall (QSH) materials, i.e., two-dimensional (2D) TIs, maintain time-reversal symmetry while resorting to spin degrees of freedom[1-3]. In solid-state materials, helical edge states possess spin-momentum locking, i.e., up and down spins

propagating in opposite directions, no matter the trivial cladding layer. Their robustness directly refers to the intrinsic spin-1/2 and fermionic time-reversal symmetry ($T_f$) of electrons, naturally guaranteeing Kramers degeneracy (Fig. 1a). In contrast, QSH in bosonic systems must construct artificial $T_f$ associated with $T_f$-related pseudospins[7-9,25-28] via interplay between bosonic spins (polarizations or modes) and bulk symmetry; however, $T_f$ is barely maintained at the boundary. Compromisingly, bosonic QSH requires trivial cladding crystals with specific symmetries or fine modifications of the boundary[29] to alleviate this mismatching[8-10]. Thus, robust transport and pseudospin-momentum locking are limited. Moreover, this mismatching is further enhanced to open a more significant gap when meeting resonators (Fig. 1b), destroying the robustness and consequently hindering its application potential for broadband topological slow waves[18-20]. Simultaneously, the same edge gap could support higher-order (HO) or spin higher-order (SHO) corner states[30,31]. The physics for the coexistence of topologically distinct QSH and HO phases remain elusive. These problems seem unsolvable in pure 2D models.

On the other hand, twisted bilayer graphene[32] intrigued the study of various bilayer systems, such as layered valley states with sites

[1]National Laboratory of Solid State Microstructures & Department of Materials Science and Engineering, Nanjing University, Nanjing 210093, China. [2]Collaborative Innovation Center of Advanced Microstructures, Nanjing University, Nanjing 210093, China. [3]Jiangsu Key Laboratory of Artificial Functional Materials, Nanjing University, Nanjing 210093, China. [4]These authors contributed equally: Xiao-Chen Sun, Hao Chen, Hua-Shan Lai. ✉e-mail: chenghe@nju.edu.cn; yfchen@nju.edu.cn

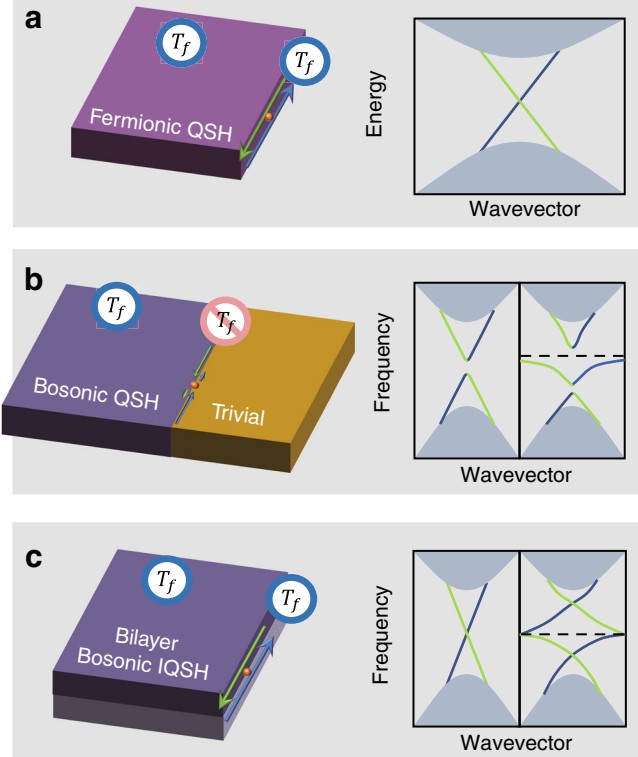

**Fig. 1 | QSH in fermionic and bosonic systems. a** Fermionic QSH case and its gapless helical edge states with $T_f$ maintained for both the bulk and boundary. **b** 2D (single-layer) bosonic QSH case with a trivial cladding crystal. There is a gap for the edge dispersion because of the absence of $T_f$. A resonator on the boundary scatters different helical states and further breaks its dispersion. **c** Bilayer bosonic IQSH case. It allows robust helical edge states without cladding crystals, which is robust against a resonator.

rotation[33], spin–Chern insulators with broken spin conservation[34], and corner states with hybrid behaviours[35]. As a result, the topological phases are vastly enriched with the extended layer degree of freedom and further advance the study of TIs, providing a new possibility for reconsidering the original boundary sensitivity of bosonic QSHs. Recently, Gladstone et al. theoretically proposed photonic SHO TI with the QSH phase via novel interactions between the transverse electric and magnetic polarizations of photons[36], yet to be realized. However, the required electromagnetic duality and imaginary coupling remain challenging in spinless airborne sound systems.

In this work, we experimentally demonstrate an acoustic ideal QSH (IQSH) in a bilayer hexagonal lattice insensitive to boundary conditions and free from cladding crystals (Fig. 1c), where 8 × 8 spinless Hamiltonian can hybridize to extract an equivalently spinful 4 × 4 Hamiltonian[2]. For the first time to our knowledge, we observe a pair of helical edge states winding several times while maintaining robustness in an unbroken time-reversal symmetry system. In this study, the IQSH acts as a topological phase transition plane that bridges the trivial, HO, and SHO phases. The entire multi-topology platform is promising for compact and tuneable devices with on-demand topological characteristics.

## Results

### Topological phase diagram

Herein, we construct a bilayer honeycomb lattice model with 120° chiral interlayer channels (Fig. 2a), acting as a Kane–Mele-like model[1] in an acoustic system with real couplings. Each unit cell contains 12 atoms (6 on each layer). The lattice constant $a$ is set to 3 cm, and the full height $H$ is $a/\sqrt{3}$. Uniform triangular prism cavities with sidelengths of $L_c = 0.4H$ and heights of $h = H/3$ represent acoustic atoms. The bilayer structure provides more symmetries than the 2D single-layer case[8] for manipulation, such as out-of-plane rotation $\hat{C}_2$, mirror, and inversion. In our structure, the chiral channels have broken mirror and inversion symmetries, while $\hat{C}_2$ could be either unbroken or broken. Intuitively, although keeping the in-plane $C_{6v}$ symmetry as that in a single-layer honeycomb lattice[8], the bilayer model may contribute excellent

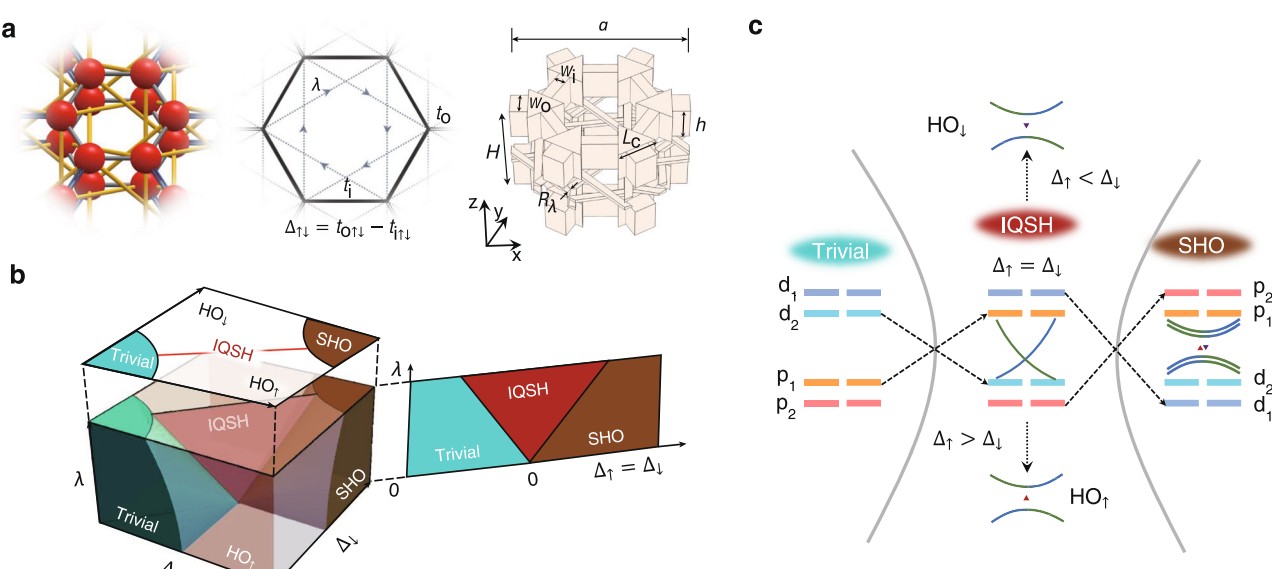

**Fig. 2 | Topological phase diagram. a** Ball-stick schematic, tight-binding model and acoustic structure of a bilayer hexagonal unit cell. $t_o$, $t_i$ and $\lambda$ represent intercell, intracell and interlayer couplings, respectively. **b** Topological phase diagram in parameter space spanned by $\Delta_\uparrow$, $\Delta_\downarrow$, and $\lambda$. $\Delta_{\uparrow\downarrow} = t_{o\uparrow\downarrow} - t_{i\uparrow\downarrow}$ is the intralayer hopping difference where the subscript arrow denotes the upper or lower layer. **c** Phase transition process relying on either bulk or edge band inversion, where $p_{1,2}$ and $d_{1,2}$ represent degenerate dipole and quadrupole modes, respectively.

topological phenomena due to the additional degrees of freedom introduced.

There are three structural parameters left for manipulation: intercell hopping (denoted as $w_o$-width tube), intracell hopping ($w_i$), and chiral interlayer coupling ($R_\lambda$), corresponding to $t_o$, $t_i$, and $\lambda$ in the tight-binding model, respectively. For simplicity and without loss of generality, we consider total intralayer hopping (intercell and intracell) in each layer to remain invariant $t_o + t_i = 2$, while their intralayer hopping difference $\Delta = t_o - t_i$ plays a pivotal factor in realizing nontrivial topology. Both $t_o$ and $t_i$ are real and positive. In a single-layer system, $\Delta > 0$ indicates larger intracell hopping, which used to be the sole parameter in realizing QSH[8]. However, the bilayer system is more complicated with additional degrees of freedom, whose Hamiltonian can be written as follows[34,36]:

$$H = -\sum_{\langle m,n\rangle_o,\alpha} t_{o\alpha}\hat{c}^\dagger_{m\alpha}\hat{c}_{n\alpha} - \sum_{\langle m,n\rangle_i,\alpha} t_{i\alpha}\hat{c}^\dagger_{m\alpha}\hat{c}_{n\alpha} - \frac{\lambda}{3}\sum_{\langle\langle m,n\rangle\rangle,\alpha\neq\beta} v_{mn,\alpha}\hat{c}^\dagger_{m\alpha}\hat{c}_{n\beta}.$$

(1)

In Eq. (1), $\hat{c}_{n\alpha}$ ($\hat{c}^\dagger_{n\alpha}$) are annihilation (creation) operators on site $n = 1, \ldots, N$, where $\alpha = \uparrow\downarrow$ denote the upper and lower layers. The first two terms represent the nearest intercell hopping and intracell hopping, respectively. The third term represents the next-nearest interlayer coupling with the coefficient $v_{mn,\alpha} = [\varepsilon_\alpha(\hat{e}_{lm}\times\hat{e}_{nl})_z + 1]/2$. Here, $\varepsilon_{\uparrow\downarrow} = \pm 1$, $m$ and $n$ indicate the next-nearest site sharing the same nearest site $l$, and $\hat{e}_{lm}$ is a unit vector pointing from $l$ to $m$. Thus, the next-nearest couplings from the upper to lower layers show clockwise chirality.

From there, we can obtain a full topological phase diagram in a three-dimensional (3D) parameter space spanned by $\Delta_\uparrow$ (upper-layer hopping difference), $\Delta_\downarrow$ (lower-layer hopping difference), and $\lambda$ (interlayer coupling). According to different topological invariants, our model can be classified into four topological phases: trivial, IQSH, HO, and SHO (shown in Fig. 2b). The IQSH phase acts as a phase transition plane (red triangular area) only in the $\Delta_\uparrow = \Delta_\downarrow$ slice, inserting into the trivial, HO, and SHO regions[35,36]. The phase transition process determined by either bulk or edge band inversion procedures is shown in Fig. 2c.

Each unit cell of our model has 12 eigenstates. After neglecting the $|s\rangle$ and $|f\rangle$ states, four pairs of twofold degenerated eigenstates $|\psi\rangle_{\uparrow\downarrow} = |p\rangle_{\uparrow\downarrow}, |d\rangle_{\uparrow\downarrow}$ at the $\Gamma$ point hybridize as follows:

$$|\psi\rangle_M = \left(|\psi\rangle_\uparrow + (-1)^M\hat{C}_{6z}|\psi\rangle_\downarrow\right)/\sqrt{2},$$

(2)

where $M = 1,2$ indicates the mixing of states in two layers, $\hat{C}_{6z}$ is the anticlockwise rotation operator of $\pi/3$ around the $z$-axis, and $|p\rangle_M = |p_x\rangle_M, |p_y\rangle_M$ and $|d\rangle_M = |d_{x^2-y^2}\rangle_M, |d_{xy}\rangle_M$ are degenerate dipole and quadrupole modes (guaranteed by $C_{6v}$ in each layer), respectively (see Part I in the Supplementary Information). Bulk band inversion between $|p\rangle_1$ and $|d\rangle_2$ corresponds to a topological phase transition from the trivial to IQSH regions. Further inversion between $|p\rangle_2$ and $|d\rangle_1$ leads to the SHO phase. For the phase transition between HO$_{\uparrow\downarrow}$ and IQSH, edge bands rather than bulk bands experience an open−close−reopen process.

Table 1 provides details for different phases. When $\lambda^2 - \Delta_\uparrow\Delta_\downarrow > 0$, interlayer coupling dominates for mixing pseudospins for two layers[34]. At this time, when $\Delta_\uparrow = \Delta_\downarrow$, the out-of-plane dimension supports the rotation operator $\hat{C}_{2y}$, allowing us to artificially define $T_f$ as follows:

$$T_f = \hat{C}_{2y}K = -\sigma_y \otimes I_4 K,$$

(3)

where $\sigma_y$ is the Pauli matrix and $K$ is the complex conjugation operator (see Part II in the Supplementary Information). $T_f$ is maintained both in the bulk and on the boundaries, guaranteeing boundary-insensitive and cladding-crystal-free acoustic IQSH with a pair of gapless helical

**Table 1 | Parameters for various phases**

| Interlayers & Intralayers | Layers ↑ & ↓ | Phases |
|---|---|---|
| $\lambda^2 > \Delta_\uparrow\Delta_\downarrow$ | $\Delta_\uparrow = \Delta_\downarrow$ | IQSH |
| | $\Delta_\uparrow > \Delta_\downarrow$ | HO$_\uparrow$ |
| | $\Delta_\uparrow < \Delta_\downarrow$ | HO$_\downarrow$ |
| $\lambda^2 < \Delta_\uparrow\Delta_\downarrow$ | $\Delta_\uparrow, \Delta_\downarrow < 0$ | Trivial |
| | $\Delta_\uparrow, \Delta_\downarrow > 0$ | SHO |

Note: $\lambda^2 = \Delta_\uparrow\Delta_\downarrow$ is the bulk degeneracy condition and indicates phase transitions.

edge states. In contrast, in HO phases with broken $\hat{C}_{2y}$ with $\Delta_\uparrow \neq \Delta_\downarrow$, their edge states are gapped, wherein one corner state appears. Depending on whether it is located on the upper or lower layer, the HO is further divided into subphases: HO$_\uparrow$ and HO$_\downarrow$. With $\lambda^2 - \Delta_\uparrow\Delta_\downarrow < 0$, the trivial phase is located in the $\Delta_{\uparrow\downarrow} < 0$ region. However, the SHO is located in the $\Delta_{\uparrow\downarrow} > 0$ region, behaving like two isolated layers due to the weak interlayer coupling.

## From the trivial to the IQSH phases

In the experiments, we begin with a phase transition from the trivial to the IQSH phases in the $\Delta_\uparrow = \Delta_\downarrow$ vertical slice, as shown in Fig. 3a. The topological invariant spin–Chern numbers ($C_\pm$)[34,36] of these three phases are 0, $\pm 1$, and 0 for the trivial, IQSH, and SHO phases, respectively (see Part III in the Supplementary Information). The calculated bulk band structures are plotted in Fig. 3b, showing an open−close−reopen process. In the trivial phase ($w_{o\uparrow\downarrow} = 0.13$cm, $w_{i\uparrow\downarrow} = 0.33$cm; $R_\lambda = a/30$), the bulk bandgap is 7.6−9.2 KHz without edge propagation, matching the measured transmission spectra well (Fig. 3c).

The phase turns from the trivial to the IQSH phases by increasing the intralayer hopping difference and/or the interlayer coupling ($w_{o\uparrow\downarrow} = 0.29$cm, $w_{i\uparrow\downarrow} = 0.31$cm, and $R_\lambda = 7a/120$). Although there remains no bulk transmission in the bandgap, edge states appear to form a pair of gapless dispersions (Fig. 3d). The frequency window for helical edge states is 7.8−8.9 KHz with a relative bandgap width over 10%. Notably, these edge states appear without cladding crystals, and only resin plates are used as hard boundaries to avoid radiation. This boundary-insensitive characteristic comes from the fact that $\hat{C}_{2y}$ fulfilled $T_f$ is held on the boundary, which remains valid even in a truncated case (see Fig. S3 in Supplementary Information).

QSH in fermionic systems is energized by a spinful $4\times 4$ Hamiltonian, i.e., BHZ model[2], which cannot be satisfied directly with only a $4\times 4$ spinless Hamiltonian in bosonic systems due to the intrinsic differences between their spins. For our acoustic system, we start with a unit cell containing 12 sites, compress it to an $8\times 8$ Hamiltonian, and finally hybridize it to extract an equivalent spinful $4\times 4$ Hamiltonian. This implies that a higher Hamiltonian dimension can compensate for a lack of spin degree, which is essential for constructing IQSH.

## Winding of helical edge states

Although the IQSH phase requires relatively harsh conditions throughout the topological phase diagram, it contributes to the robust insensitivity of the gapless helical edge states. The experimental setup is shown in Fig. 4a. Our bilayer acoustic sample has $21\times 4\sqrt{3}$ periods in the $xy$-plane with full dimensions of $63.0\times 22.5\times 1.7$cm$^3$. In our experiments, we excite the sound propagation along the boundary and measure the amplitudes and phases at each site in 21 periods. The phases are calibrated by using an additional reference detector (see Methods and Supplementary Information for details). After Fourier transformation, we obtain the edge dispersion with a resolution of 0.1 ($\pi/a$). The measured helical edge states for intact and truncated boundaries are shown in Fig. 4b, c, respectively. The ribbon super unit cells are shown in the right panels, which are periodic in the $x$-direction and have different boundary configurations in the $y$-direction. The

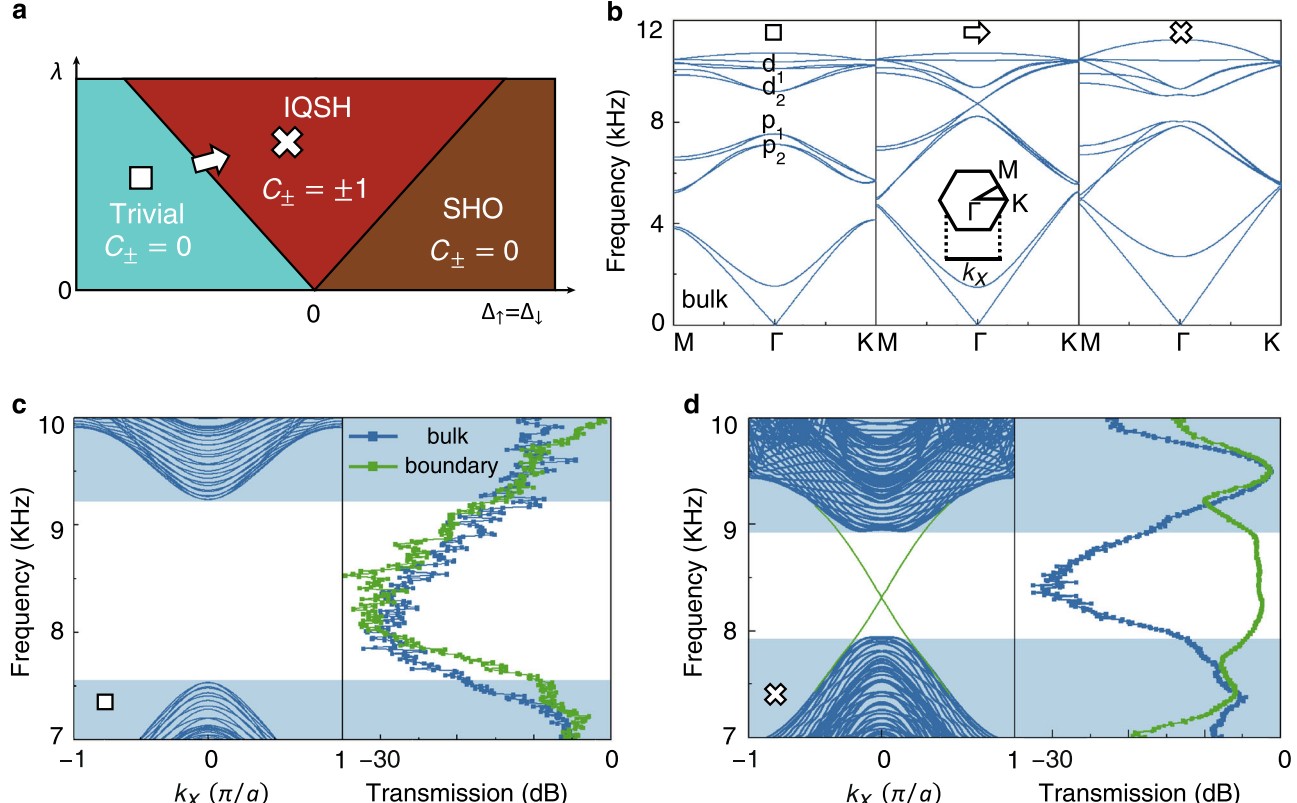

**Fig. 3 | Trivial and IQSH phases. a** Topological phase slice when $\Delta_\uparrow = \Delta_\downarrow$. Insets show spin–Chern numbers. **b** Bulk bands inversion associated with the phase transition from the trivial to the IQSH phases. **c, d** Numerical projected band structures and experimental transmission spectra for the trivial and IQSH phases, respectively. Here, only hard boundary conditions are used.

boundary decoration only slightly shifts the frequencies of the edge Dirac nodes, leaving them both gapless. The helical edge states are robust against various disorder types, including cavity, disorder, and Z-shaped corners, and they are experimentally checked (see Part IV in the Supplementary Information).

The gapless helical edge states of acoustic IQSH with boundary-insensitive properties are good candidates for realizing robust slow waves over wide frequency windows. The winding strategy for realizing topologically slow waves was proposed in photonic systems with broken time-reversal symmetry, e.g., the photonic anomalous quantum Hall effect under magnetic bias[18–20]. Due to the nonreciprocal chiral edge states, the edge dispersion winds in the Brillouin zone without opening a gap. But this scheme cannot be directly transplanted to the magnetic-free cases because the lack of $T_f$[8,9,37–41] harshly breaks the edge dispersion when meeting resonators (Fig. S10). Thus, the spin-momentum locking and the robust wave propagation are destroyed, leading to backscattering. However, our IQSH handles this problem and winds the pair of helical states to realize extremely slow yet robust edge wave propagation against backscattering throughout the bulk bandgap window.

Benefiting from the ideal QSH phase, the approach to achieving this goal is very simple. We only need to add a pendant cavity ($w_o$-width connecting tube) as a local resonator on each boundary site. As shown in Fig. 4d, the edge dispersion is flattened and winds once more; nevertheless, the spin-momentum locking and robust wave propagation are largely maintained. As a result, the sound velocity is retarded significantly. This direct and simple design leaves a tiny gap (approximately 0.27% relative to the bandgap center frequency and 2.4% relative to the bandgap width) in the winding dispersion on the boundary due to the size effect and the inevitable resonator-affecting fluctuation of pseudospins. Moreover, we can further minimize this

gap and slow the sound propagation by tuning the shapes of pendant cavities, winding it twice with flatter dispersion, slower wave and negligible tiny gap (see Part V in Supplementary Information). Note that local defects in our model must be smooth in the z-direction (identical between two layers) to fulfil $T_f$ and avoid interspin scattering.

## HO and SHO phases

The edge states open a gap when breaking the $\hat{C}_{2y}$ symmetry ($\Delta_\uparrow \neq \Delta_\downarrow$). In Fig. 5a, we display the phase diagram in the $\lambda = 1$ horizontal slice. The edge dispersion gaps, wherein the corner state, e.g., HO phase, can appear. In our bilayer model, the layer components $\uparrow\downarrow$ are coupled through $\lambda$, making the corner charge $Q_c^{\uparrow\downarrow}$ for each layer no longer quantized as in single-layer cases. Here, the modified corner charges are defined by weighing the layer contribution for low-energy bands as follows[36,42,43]:

$$Q_c^{\uparrow\downarrow} = \frac{1}{4}\sum_i \left( |\langle \uparrow\downarrow | \psi_i(M) \rangle|^2 - |\langle \uparrow\downarrow | \psi_i(\Gamma) \rangle|^2 \right). \quad (4)$$

where $|\psi_i(\mathbf{k})\rangle$ is the low-energy wave function at the $\mathbf{k} = M, \Gamma$ points. Then, the total spin corner charge can be quantized as $Q_c = Q_c^\uparrow + Q_c^\downarrow = 1/2$, leading to only one corner state in this phase (see Part VI in the Supplementary Information). For convenience, we define the spin imbalance $S_c = Q_c^\uparrow - Q_c^\downarrow$ and divide the HO phase into $HO_\uparrow$ ($S_c > 0$) and $HO_\downarrow$ ($S_c < 0$), in which their acoustic field distributions are mainly located on the upper and lower layers, respectively. Note that the $HO_\uparrow$ and $HO_\downarrow$ phases acting as a $\hat{C}_2$-connecting pair are intrinsically the same, sharing the same topological charge.

To measure the HO phase, a diamond-shaped sample of 8 × 8 unit cells marked with pump-probe configurations is shown in Fig. 5b. The

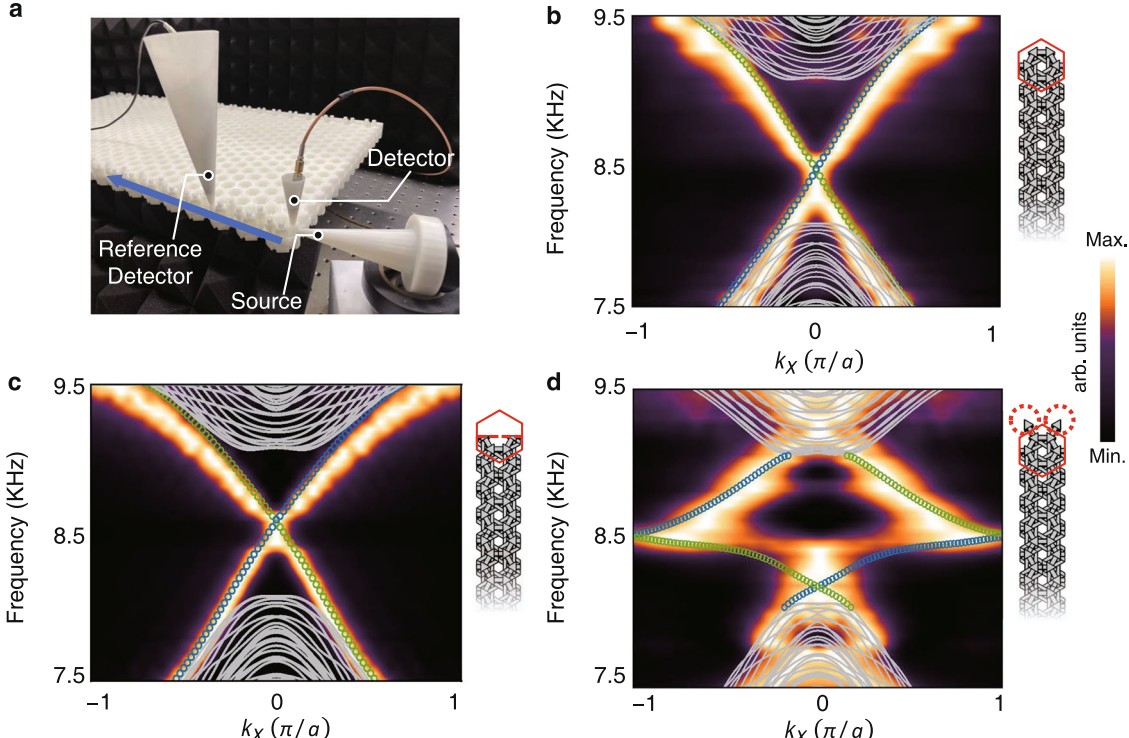

**Fig. 4 | Robustness of ideal helical edge states. a** Experimental setup to measure the dispersion. The blue arrow marks the acoustic propagation. **b** Experimentally measured band structures with intact boundary along the *x*-direction. The colour scale represents the strength of the acoustic energy density with arbitrary units (arb. units). The grey lines denote the calculated bulk bands. The blue (green) circles denote calculated acoustic pseudospin up (down) edge states. **c** Truncated case with half unit cells. **d** Winding once for acoustic helical edge states when coupled to boundary pedant resonators.

parameters are $w_{o\uparrow} = 0.4$ cm, $w_{o\downarrow} = 0.13$ cm, $w_{i\uparrow} = 0.2$ cm, $w_{i\downarrow} = 0.33$ cm, and $R_\lambda = a/30$. The experimental response spectra clearly show a bulk gap of 7.3−9.4 KHz and an edge gap of 7.5−8.1 KHz, wherein the corner state is located at approximately 7.8 KHz with a 25 dB peak (Fig. 5c). The experimental data are in good agreement with the numerical results. Our fabricated sample is in the HO$_\uparrow$ phase, and one can find its $\hat{C}_2$-connecting partner in the HO$_\downarrow$ phase.

The phase transitions from the HO phase to the SHO phase by increasing the lower intralayer hopping difference ($w_{o\downarrow} = 0.4$ cm, $w_{i\downarrow} = 0.2$ cm) while maintaining the other parameters. Unlike the HO case, the SHO corner charges for two layers are separately quantized as $Q_c^\uparrow = Q_c^\downarrow = 1/2$ according to Eq. (4), thus supporting two corner states with different pseudospins. As shown in Fig. 5d, the SHO case shows a bulk bandgap of 7.5−9.3 KHz and an edge bandgap of 7.6−8.3 KHz, in which we find two types of corner states located at approximately 7.9 KHz with different symmetries, as shown in Fig. 5b and Fig. 5e. The symmetric S-mode shows the symmetric phase relative to the *xy* plane, while the AS-mode shows the anti-symmetric phase (see Methods and Supplementary Information for detailed experimental setup). These two corner states originate from two pairs of gapped edge states due to the thorough band inversions for four paired p−d bulk bands. Note that the slightly different frequencies between them are due to the finite-size effect[36].

## Discussion

The full topological phase diagram obtained in a high-dimensional parameter space offers an opportunity to re-examine pure 2D QSH phases. Our 3D topological phase diagram collapses into a line in the 2D limitation, i.e., the $\lambda = 0$ line on the $\Delta_\uparrow = \Delta_\downarrow$ plane (Fig. 2b). There is only one crucial point for IQSH, i.e., the phase transition point between trivial and HO. From this aspect, we can clearly understand the physics behind why the previous pure 2D QSH cases are boundary sensitive

with edge gaps wherein HO coexists[8,31]. However, in our bilayer structure, the IQSH point broadens to a phase transition plane connecting two HO phases, guaranteeing gapless helical edge states and robustness against imperfections. Compared to the recently proposed photonic bilayer model[36], our acoustic structure only requires one polarization and real couplings, which is convenient for experiments. Note that if we release more degrees of freedom, such as by breaking $C_{6v}$, various topological valley states are possible[33]. Furthermore, there are alternative choices for the $\hat{C}_2$ operator in the *xy* plane associated with fermionic $T_f$. Thus, our IQSH phase remains valid for various boundary conditions, including zigzag and armchair.

To conclude, in a time-reversal invariant bilayer acoustic system, we experimentally demonstrate abundant topological phases, in which IQSH plays the role of the neighbouring and parent phase for realizing other phases. Our IQSH releases boundary restriction and serves as a good candidate for more robust and compact topological devices. The observed flat yet robust polariton-like helical edge dispersion (winding several times) makes significant progress for achieving waves that are extremely slow and robust over a wide frequency window (exceeding 10%), which has long been sought after in industrial delay devices. This design strategy can be readily extended to other systems, including photonics. Notably, the strengthened interactions between slow waves and matter may advance future topological insulator lasers[22] and quantum emitters[44] with smaller sizes and higher performance levels. Moreover, this model can serve as a multifunctional platform to reveal other topological phases and phase transitions by considering more degrees of freedom, such as disorder in a topological Anderson TI[45], gain/loss in a non-Hermitian model[46,47], or braiding in a non-Abelian system[48,49]. The entire multi-topology platform investigated in higher-dimensional parameter space may shed light on tuneable devices with on-demand topological characteristics.

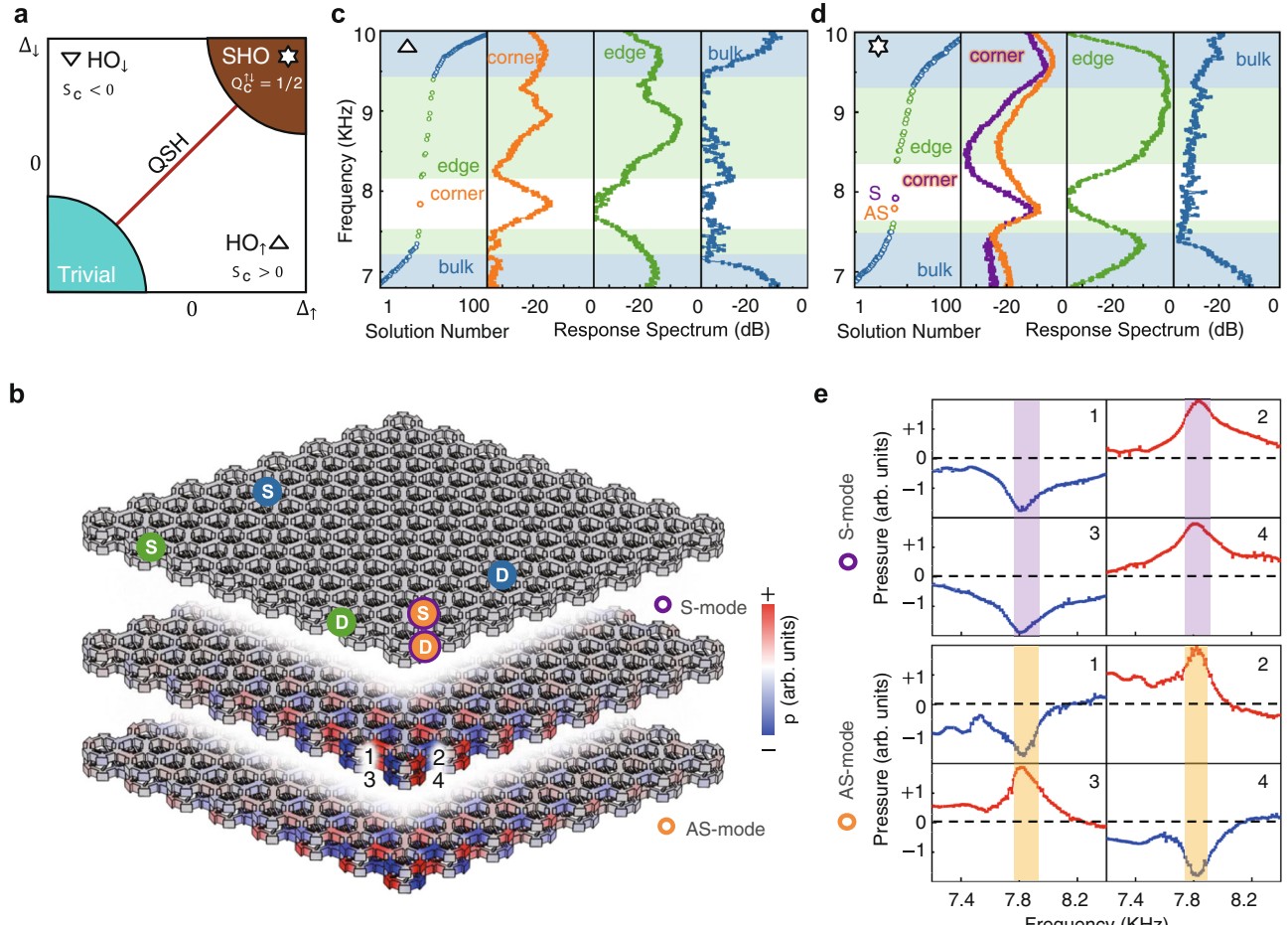

**Fig. 5 | Response spectra for HO and SHO phases. a** Phase diagram at $\lambda = 1$ slice. The insets show corner charges. **b** Sample structure for measurements (upper panel). The blue, green, and orange markers S (D) denote the bulk, edge, and corner sites, respectively, for the placement of acoustic sources (detectors) in experiments. The middle (lower) panel shows the simulated field distribution for the S-mode (AS-mode) in the SHO phase, with colour scale representing the strength of the acoustic pressure with arb. units. **c** Measured HO phase, including response spectra for the corner, edge and bulk versus the calculated results (left column). **d** Measured SHO phase case. **e** Measured acoustic pressure with arb. units. for S- and AS-modes at sites 1–4 marked in b.

## Methods

### Experimental measurement

All samples used in the experiments are fabricated using photosensitive resin (Godart[TM] 8111X) via 3D printing (geometry tolerance of 0.1 mm). This stereolithography material (modulus 3160 MPa, density 1.14 gcm$^{-3}$) is regarded as an acoustic hard boundary for the impedance mismatch. The thicknesses of all sites and tube walls are 1 mm. In the experiments, we drill some equilateral triangle holes with side-lengths of $0.2\sqrt{3}$ cm on the cavities at the upper and lower surfaces and print a corresponding plug for the convenience of measurements (see Fig. S5 in the Supplementary Information for details).

In the measurements, commercial loudspeakers (AMT-47) and microphones (BSWA MPA416) are used as the acoustic source and detector, respectively. In Fig. 4, 21 points from each unit cell are detected under a frequency sweep utilizing the designed holes. Both the amplitude and phase information are collected with NI cDAQ-9185. We use an additional reference detector to calibrate the phase at each site to reduce errors. For the response spectra in Fig. 5c, d, the positions of the sources and detectors can be seen from Fig. 5b marked by S and D. Note that for the measurements of corner states of the SHO phase in Fig. 5e, a pair of speakers are used at the orange–purple site S shown in Fig. 5b, with one speaker at the top surface and the other at the bottom surface of the sample, respectively. The relative phases of these

two speakers are controlled to generate corner states with different symmetries. Four sites at the corner are detected, marked as 1, 2, 3, and 4 in Fig. 5b, e.

### Numerical simulation

Full-wave simulations are implemented by the commercial software COMSOL Multiphysics with a 3D acoustic module based on a finite element method. The mass density and sound velocity are 1.21 kgm$^{-3}$ and 343 ms$^{-1}$, respectively. When calculating the bulk band in Fig. 3b, the periodic boundary condition is used in the $xy$-directions with a hard boundary for the $z$-direction. When calculating the boundary band in Figs. 3c, d and 4b–d, only the $x$-direction boundary is set as a periodic condition. All boundaries are hard for corner states in Fig. 5c, d.

## Data availability

The simulated and experimental data are under private user license which cannot be made public. The data that support the plots within this paper and the other findings of this study are available from the corresponding authors upon reasonable request.

## Code availability

All related codes can be built with the instructions in the Supplementary Information and available from the corresponding authors upon reasonable request.

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

## Acknowledgements

The work was jointly supported by the National Key R&D Programme of China [Grant No. 2022YFA1404302 (C.H.)] and the National Natural Science Foundation of China [Grant Nos. 52022038 (C.H.), 92263207 (C.H.), 11874196 (C.H.), 11890700 (Y.-F.C.), 51721001 (Y.-F.C.), 52027803 (Y.-F.C.), and 52103341 (X.-C.S.)]. We thank Dr. Ze-Guo Chen and Si-Yuan Yu for providing helpful discussions.

## Author contributions

X.-C.S. and C.H. conceived the original idea. X.-C.S., H.-S.L., and C.H. performed the theoretical portions of this work. H.C., C.-H.X., and C.H. conducted the experiments. C.H. and Y.-F.C. supervised the project. All authors contributed to the analyses and the preparation of the paper.

## Competing interests

The authors declare no competing interests.
