## [Peer review file · Nature Communications]

Ideal acoustic quantum spin Hall phase in a multi-topology platformEditorial Note: This manuscript has been previously reviewed at another journal that is not operating a transparent peer review scheme. This document only contains reviewer comments and rebuttal letters for versions considered at *Nature Communications*.

REVIEWERS' COMMENTS:

Reviewer #2 (Remarks to the Author):

The response letter has successfully addressed all my concerns raised in the last report. The new experimental results are very nice, which clearly support their analysis. I am happy to recommend it for publication in Nature Communication.

Reviewer #3 (Remarks to the Author):

I have re-read the manuscript, as well as the authors responses to my critiques. Those have been adequately answered. The English has also been improved significantly. Additional experiments demonstrating the robustness of the edge states have also been adequate. Therefore, I recommend the publication of this paper.

Reviewer #2--- NCOMMS-22-44987-T

Comment of Reviewer #2:

The response letter has successfully addressed all my concerns raised in the last report. The new experimental results are very nice, which clearly support their analysis. I am happy to recommend it for publication in Nature Communication.

Our response: We appreciate Reviewer #2 for their positive recommendation.

Reviewer #3--- NCOMMS-22-44987-T

Comment of Reviewer #3:

I have re-read the manuscript, as well as the authors responses to my critiques. Those have been adequately answered. The English has also been improved significantly. Additional experiments demonstrating the robustness of the edge states have also been adequate. Therefore, I recommend the publication of this paper.

Our response: We thank Reviewer #3 for their recommendation.